# AI-driven discovery of novel extracellular matrix biomarkers in pelvic organ prolapse

Yanlin Mi[1,2]*, Ben Cahill[1], Venkata V. B. Yallapragada[3], Reut Rotem[4,5], Barry A. O'Reilly[4], Sabin Tabirca[1,6]

1 School of Computer Science and Information Technology, University College Cork, Cork, Ireland, 2 SFI Centre for Research Training in Artificial Intelligence, University College Cork, Cork, Ireland, 3 Centre for Advanced Photonics and Process Analytics, Munster Technological University, Cork, Ireland, 4 Department of Urogynaecology, Cork University Maternity Hospital, Cork, Ireland, 5 Department of Obstetrics and Gynecology, Shaare Zedek Medical Center, Hebrew University School of Medicine, Jerusalem, Israel, 6 Faculty of Mathematics and Informatics, Transylvania University of Brasov, Brasov, Romania

* y.mi@cs.ucc.ie

## Abstract

Deep learning for protein function prediction faces significant challenges in identifying disease-specific proteins. We present Extracellular Matrix Protein Predictor (EPOP), an advanced transfer learning framework leveraging protein language models to decode disease mechanisms. Focusing on pelvic organ prolapse (POP), which affects up to 50% of women worldwide, EPOP demonstrates AI's power to reveal novel therapeutic targets. We developed a sophisticated fine-tuning protocol for the ESM-2 model, optimized for ECM protein prediction. Our architecture integrates specialized attention mechanisms with interpretability modules, trained on expertly curated and balanced datasets totaling 80,000 proteins (40,000 ECM and 40,000 non-ECM). The framework employs a novel validation strategy using a 16,000-sample independent test set and clinical proteomics data. EPOP achieved unprecedented performance (99.40% accuracy) in ECM protein classification, significantly surpassing traditional deep learning architectures (10.81% improvement over Transformer models, 21.71% over Long Short-Term Memory). Applied to clinical samples, our model revealed a previously unknown pattern of ECM remodeling, identifying 24 novel disease-associated proteins. Model interpretability analysis uncovered specific sequence motifs and structural features critical for ECM protein function, providing mechanistic insights into disease progression. EPOP demonstrates how advanced AI bridges molecular analysis and clinical applications, uncovering novel therapeutic targets. Its success suggests broader applications across ECM-related disorders, potentially transforming approaches to diseases affecting connective tissue architecture.

**Data availability statement:** All datasets, trained models, and source code used in this

study are publicly available on GitHub at: https://github.com/Atobelin/EPOP. The repository includes: 1. Datasets: Curated extracellular matrix (ECM) and non-ECM protein sequences used for training and validation. 2. Code: Python scripts for data preprocessing, model training, evaluation, and interpretability analysis. 3. Model: The fine-tuned EPOP model ready for application to new datasets. Additionally, the mass spectrometry-based proteomic dataset used for external validation is publicly available via the PRIDE Archive under accession number PXD011467. For questions regarding data reuse or institutional permissions, please contact the Computer Resources for Research group at University College Cork (https://crr.ucc.ie/).

**Funding:** This publication has emanated from research conducted with the financial support of Taighde Éireann – Research Ireland under Grant No. 18/CRT/6223 awarded to Y.M. She received a full PhD scholarship through this program. The funders provided financial support for the author's time but had no role in study design, data collection and analysis, decision to publish, or preparation of the manuscript.

**Competing interests:** The authors have declared that no competing interests exist.

## Author summary

The extracellular matrix forms the structural scaffold of tissues throughout the body, crucial for maintaining organ support and function. Disruptions in this complex protein network contribute to various disorders, including POP, a condition significantly impacting quality of life and affecting up to half of women worldwide. Identifying specific proteins involved in POP has been challenging with traditional methods. We addressed this by developing EPOP, an AI system leveraging protein language models to predict disease-associated proteins from amino acid sequences. Through a sophisticated two-stage transfer learning approach, our AI model achieved 99.40% accuracy in identifying extracellular matrix proteins. Applied to clinical samples, EPOP uncovered a previously unrecognized pattern of tissue remodeling and identified 24 novel disease-associated proteins, such as POSTN. Our interpretability analysis revealed specific sequence features crucial for protein function, providing new insights into disease mechanisms. This methodology demonstrates how advanced AI can transform biomarker discovery in complex disorders and holds promise for application across numerous extracellular matrix-related conditions.

## Introduction

Recent advancements in artificial intelligence (AI) have transformed biomarker discovery across medical fields [1]. Machine learning (ML), particularly Deep Learning (DL), now identifies disease-specific biomarkers inaccessible to conventional methods [2,3]. For example, breast cancer studies leveraging recursive feature elimination revealed nine diagnostic biomarkers (e.g., COL10A1 and S100P), validated by qRT-PCR with an AUC of 0.989 [4]. And integrating multi-omics data through XGBoost classifiers uncovered cardiovascular biomarkers (RPL36AP37, HBA1) with 100% predictive accuracy [5], while metabolomic analysis via logistic regression outperformed traditional amyloid-beta detection in Alzheimer's diagnosis using choline and lactate levels [6]. Protein language models further exemplify AI's impact, ESM-2 [7] and transformer architectures [8] enable high-throughput structure-function analysis, culminating in tools like ESM Metagenomic Atlas [7] and AlphaFold 2 [9], that achieve near-experimental prediction accuracy. These AI-driven approaches are particularly valuable for complex disorders, offering screening of thousands of potential biomarkers simultaneously.

Pelvic organ prolapse (POP) represents a significant health challenge, affecting millions of women worldwide with a lifetime risk of surgery estimated at 12.6% to 19% [10]. This disorder, characterized by the descent of pelvic organs into the vaginal canal, not only impairs quality of life but also poses substantial economic burdens on healthcare systems [11]. Despite its prevalence and impact, the molecular underpinnings of POP remain elusive, hampering the development of effective diagnostic and therapeutic strategies [12].

At the heart of POP pathogenesis lies the extracellular matrix (ECM), a complex network of proteins, glycoproteins, and proteoglycans that forms the structural scaffold of pelvic floor supportive tissues [13,14]. The ECM's role extends beyond mere structural support; it orchestrates cellular behavior, tissue homeostasis, and biomechanical properties crucial for pelvic organ support [15]. Mounting evidence suggests that disruptions in ECM metabolism and organization are pivotal in POP development, manifesting as alterations in collagen composition, elastin degradation, and dysregulation of matrix metalloproteinases [16,17].

However, systematically unraveling the complex web of ECM proteins involved in POP has proven to be challenging. Traditional experimental approaches, while valuable, are often limited by their throughput, bias toward known proteins, and inability to capture the full spectrum of ECM alterations [18]. The complex nature of the ECM, with its diverse protein components and dynamic remodeling processes, further complicates comprehensive analysis using conventional methods [19]. Additionally, the heterogeneity of pelvic support tissues presents analytical challenges that can obscure subtle but significant alterations in ECM composition and organization [20–23]. This knowledge gap calls for innovative strategies to comprehensively map the ECM landscape in POP and identify novel therapeutic targets with potential clinical applications [7].

The advent of artificial intelligence, particularly protein language models like the Evolutionary Scale Modeling 2 (ESM-2), offers a transformative approach to these challenges [7]. Based on Transformer architecture, ESM-2 is trained on vast, evolutionarily diverse protein sequence databases through self-supervised learning to capture complex amino acid contexts and intricate patterns within protein sequences [24,25]. The generated representations integrate both sequence conservation patterns and implicit structural features, enabling high-throughput unbiased prediction of protein functions with unprecedented precision, approaching experimental precision in certain contexts [26,27]. With its innovative architecture that integrates structural priors, ESM-2 represents a significant leap forward in our ability to decode protein sequences and infer functional implications [28,29].

In this study, we used the power of ESM-2 to develop a new computational framework, the ECM Protein Predictor for Pelvic Organ Prolapse (EPOP), which aims to systematically identify and characterize POP-associated ECM proteins (Fig 1). By fine-tuning ESM-2 on expertly curated datasets of ECM and non-ECM proteins, we create a highly accurate tool capable of predicting ECM functionality from amino acid sequences alone. We then apply

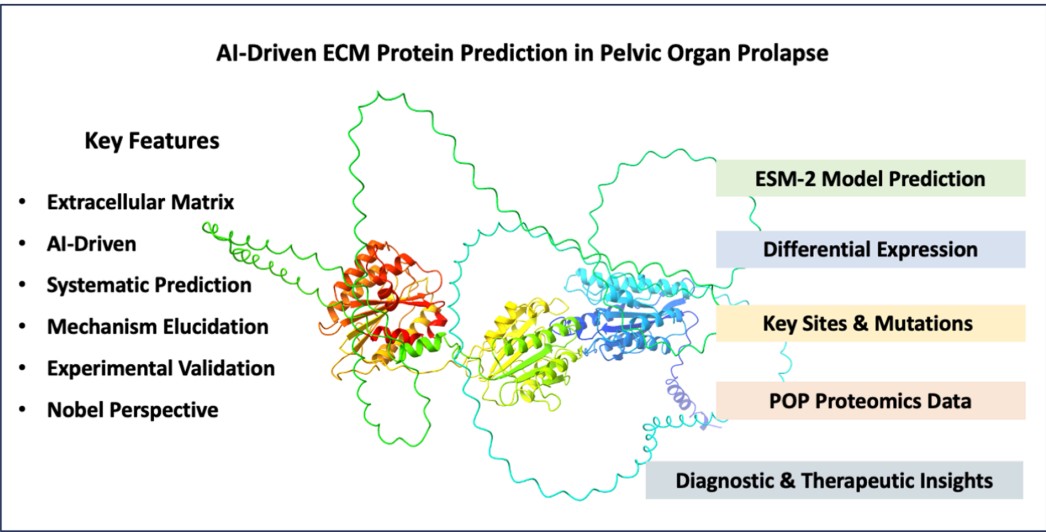

**Fig 1. AI-driven ECM protein prediction in POP.** The figure illustrates the key components of our analytical framework. Left panel lists six key features of the approach: Extracellular Matrix, AI-Driven, Systematic Prediction, Mechanism Elucidation, Experimental Validation, and Novel Perspective. Center shows a protein structure visualization with multiple domains in different colors. Right panel displays five key analytical outputs: ESM-2 Model Prediction, Differential Expression, Key Sites & Mutations, POP Proteomics Data, and Diagnostic & Therapeutic Insights, each represented by a distinctive icon and box style.

EPOP to analyze proteomics data from POP patient samples, uncovering potential key ECM proteins and evaluating their known associations with POP pathogenesis through literature cross-validation and functional enrichment.

Our approach not only deepens our understanding of the ECM-mediated molecular pathogenesis of POP but also demonstrates the potential of AI-driven strategies in uncovering novel disease-associated proteins. By bridging the gap between large-scale protein sequence data and disease-specific molecular mechanisms, we pave the way for more targeted experimental investigations and potential therapeutic interventions. Moreover, the EPOP framework holds promise for extension to other ECM-related disorders, offering a versatile tool for the systematic discovery of pathogenic mechanisms and therapeutic targets across a spectrum of diseases.

This study represents a significant step forward in leveraging AI to decode the complexities of the ECM in disease contexts, potentially revolutionizing our approach to understanding and treating POP and related disorders. As we unravel the intricate relationships between ECM composition, function, and POP pathogenesis, we open new avenues for personalized medicine and targeted therapies in women's health.

## Materials and methods

Our study employed a comprehensive workflow (Fig 2) integrating data curation, advanced machine learning techniques, and biological validation to predict and characterize POP-associated ECM proteins.

### Data collection and preprocessing

**ECM and non-ECM protein datasets.** We curated two high-confidence datasets from UniProt [30] for model training and evaluation, resulting in a balanced dataset of 80,000 proteins (40,000 ECM and 40,000 non-ECM) to ensure class balance and computational efficiency.

**Positive dataset (ECM proteins):** We initially retrieved 40,155 ECM-related proteins annotated with the Gene Ontology (GO) [31] term "extracellular matrix organization" (GO:0031012) and its descendant terms [31]. While this GO term broadly captures biological processes involved in ECM dynamics, not all annotated entries represent true ECM components. To refine the dataset, we applied a rigorous two-step curation strategy. Computational biologists filtered proteins based on reviewed annotation status and high-confidence experimental evidence codes. Then, an experienced cell biologist manually evaluated each candidate using UniProt functional descriptions and supporting literature. Proteins were retained only if they demonstrated clear extracellular localization and experimentally supported roles in ECM structure or regulation. Proteins with exclusively intracellular localization, membrane-bound proteins without ECM function, or non-secreted factors unrelated to matrix organization were excluded. From this curated set, we randomly selected 40,000 proteins to constitute the positive class for downstream modeling.

**Negative dataset (Non-ECM proteins):** We initially collected 46,427 proteins from UniProt that lacked any ECM-related GO annotations. To further minimize the inclusion of overlooked ECM components, we applied an additional two-step filtering procedure. SignalP [32] was used to exclude proteins predicted to contain signal peptides, which often indicate secretion into the extracellular space. Then, TMHMM [33] was used to remove proteins with predicted transmembrane domains, thereby eliminating integral membrane proteins that may interact with the ECM but are not themselves its structural components. From this filtered

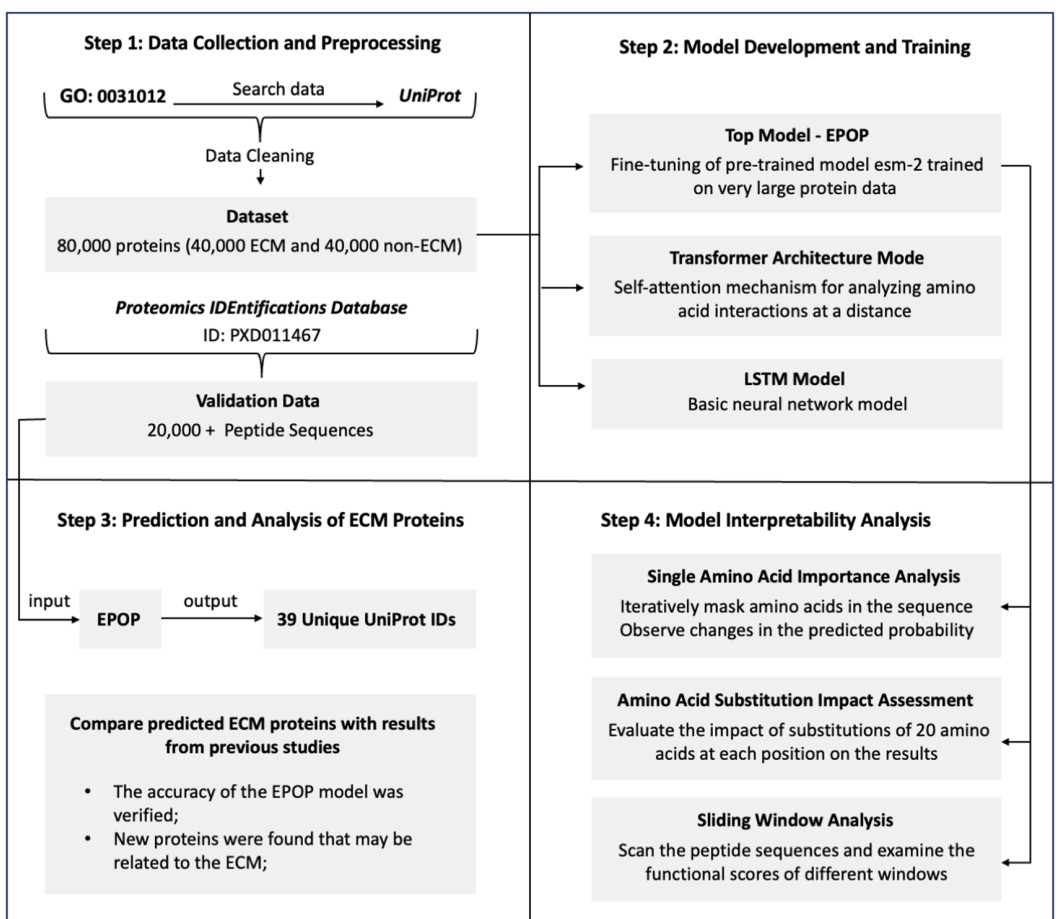

**Fig 2. Workflow overview showing the four main steps of the EPOP methodology.** Step 1 (Data Collection and Pre-processing) shows the retrieval of data from GO:0031012 and UniProt. Initially, this resulted in raw datasets of 40,155 ECM-related proteins and 46,427 non-ECM proteins. These were subsequently cleaned, balanced to 40,000 ECM and 40,000 non-ECM sequences, and then split into a training set (48,000 sequences), a validation set (16,000 sequences), and a test set (16,000 sequences) for model training. Additionally, proteomic data from PRIDE project ID PXD011467, comprising 22,341 PSMs from patient samples, was utilized for external validation and application. Step 2 (Model Development and Training) illustrates three parallel models: EPOP (fine-tuning of pre-trained ESM-2), Transformer Architecture Model (self-attention mechanism), and LSTM Model (basic neural network). Step 3 (Prediction and Analysis) shows EPOP's input-output process. Specifically, the trained EPOP model was applied to the PRIDE proteomic data, identifying 40 unique UniProt IDs of proteins predicted as ECM based on the optimal classification threshold, which will be further analyzed and validated against existing literature. Step 4 (Model Interpretability Analysis) details three analysis approaches: Single Amino Acid Importance Analysis, Amino Acid Substitution Impact Assessment, and Sliding Window Analysis, which were applied to the high-scoring peptides to identify key functional regions and residues.

pool, we randomly selected 40,000 proteins to constitute the negative class for downstream modeling.

To reduce sequence redundancy and enhance dataset diversity, we applied CD-HIT [34] with a 30% sequence identity threshold to both datasets. This threshold is a widely accepted heuristic in protein bioinformatics for balancing redundancy reduction with the preservation of distinct biological information, allowing us to group likely homologous sequences while retaining a diverse representation of protein families. By removing highly similar sequences, this step mitigates the risk of model overfitting to specific sequence variations and ensures

that the model learns more generalizable features indicative of ECM or non-ECM characteristics. This approach results in a dataset that facilitates robust model training and provides a more reliable evaluation of classification performance by preventing artificial inflation of metrics due to highly similar sequences in training and test sets.

In the end, the comprehensive balanced dataset of 80,000 proteins (40,000 ECM and 40,000 non-ECM) was randomly partitioned into three mutually exclusive sets: a training set of 48,000 sequences, a validation set of 16,000 sequences, and an independent test set of 16,000 sequences. This random partitioning ensured that each subset was representative of the overall protein diversity and that the test set remained completely unseen during model training and validation phases, thereby allowing for an unbiased assessment of generalization performance.

**POP-related proteomic data.** We obtained proteomic data comparing vaginal wall tissues of POP patients and healthy controls from Li et al. [63] (PRIDE [35]: PXD011467). Raw mass spectrometry data were processed using MaxQuant [36] with the following parameters:

1. False discovery rate (FDR) threshold: 1% at both peptide and protein levels
2. Label-free quantification (LFQ) for protein abundance estimation

To ensure data reliability, we implemented comprehensive quality control measures (Fig 3):

1. Peptide length distribution analysis
2. Percolator posterior error probability (PEP) evaluation
3. Correlation analysis between Mascot scores and PEP values

## Model development and training

**EPOP model: fine-tuning ESM-2.** We leveraged the state-of-the-art ESM-2 protein language model [7] as the foundation for our ECM function prediction task. Specifically, we

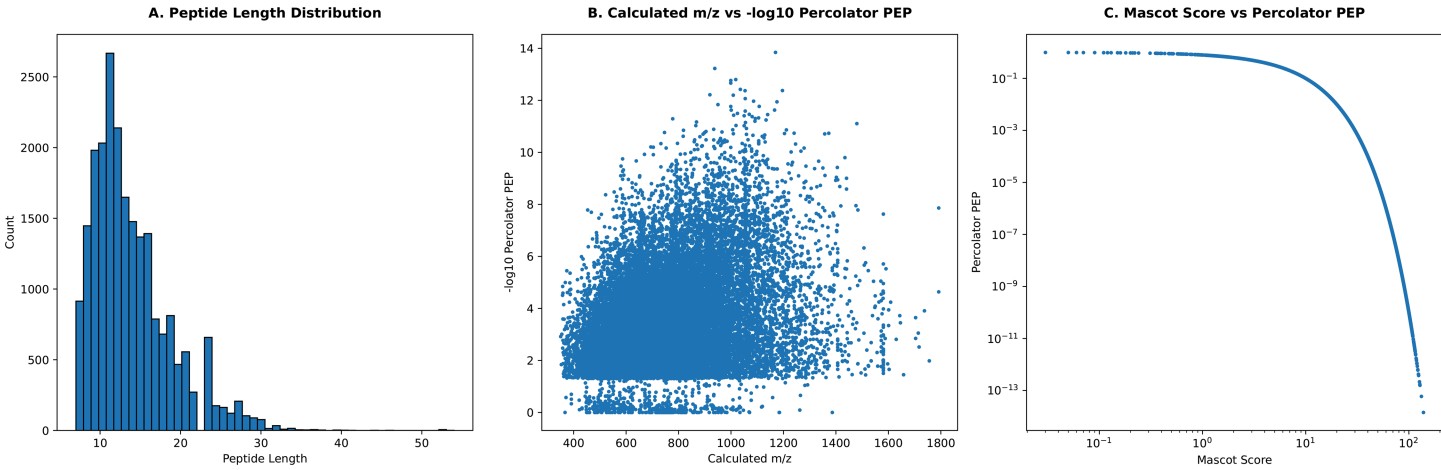

**Fig 3. Quality control metrics for POP proteomic data. A:** Peptide length distribution histogram showing the frequency distribution of peptide lengths, with peak frequency around 10-15 amino acids. **B:** Scatter plot showing the relationship between calculated m/z values (x-axis, 400-1800) and -log10 Percolator PEP scores (y-axis, 0-14), displaying a broad distribution of peptide identifications. **C:** Scatter plot demonstrating the correlation between Mascot scores (x-axis, log scale) and Percolator PEP values (y-axis, log scale), showing a strong negative correlation.

used the pre-trained ESM-2 (650M parameters, 33 layers, UR50D) checkpoint from Meta AI, which provides a powerful foundation with a large parameter count for complex biological sequence modeling.

Our fine-tuning process employed a two-stage transfer learning approach on the prepared training dataset (48,000 sequences):

1. Initial Adaptation
   - Dataset: The entire training set (48,000 sequences) was used for this stage.
   - Training parameters: The classification head was trained for 4 epochs.
   - Batch size: 32.
   - Optimizer: AdamW [37] (learning rate: 2e-5, weight decay: 0.01).
   - Objective: This stage aimed to quickly adapt the newly added classification layer to the ECM prediction task while keeping the pre-trained ESM-2 layers frozen.
2. Full Fine-Tuning
   - Dataset: The entire training set (48,000 sequences) was again used.
   - Training parameters: The entire ESM-2 model (including pre-trained layers and the classification head) was fine-tuned for 2 epochs.
   - Batch size: 8 (with a gradient accumulation of 2, resulting in an effective batch size of 16).
   - Optimizer: AdamW (learning rate: 1e-5, weight decay: 0.02).
   - Objective: This stage allowed for more subtle adjustments across all layers of the pre-trained model, enabling it to capture highly specific features relevant to ECM protein classification.

The selection of these training parameters (epochs, batch size, AdamW optimizer with its learning rate and weight decay) was guided by empirical exploration. This approach aimed to mitigate catastrophic forgetting and prevent overfitting, common challenges when adapting large pre-trained models like ESM-2 to a specific classification task. While a formal hyper-parameter optimization sweep was not conducted, these settings were carefully determined through preliminary experimentation to achieve satisfactory convergence and ensure the reproducibility of our fine-tuning process.

Throughout both stages, model performance was continuously monitored on a held-out validation set of 16,000 samples to assess convergence and generalization ability. The model checkpoint achieving the best performance on the validation set was saved for final evaluation.

**Baseline models.** To benchmark EPOP's performance, we developed two baseline deep learning models, both trained on the same balanced dataset (training set: 48,000 sequences, validation set: 16,000 sequences).

1. Transformer model
   - Architecture: Our custom Transformer model consisted of a single transformer encoder layer with 8 attention heads and a hidden dimension (embedding size) of 128. Sequence tokenization used a vocabulary size of 27 (20 standard amino acids plus special tokens).
   - Input: Tokenized amino acid sequences.
   - Output: Binary classification (ECM-related or not) via a final dense layer.
2. LSTM model
   - Architecture: This model featured a Bidirectional LSTM layer with 32 hidden units, followed by a dense classification layer.

- Input: Same tokenized amino acid sequences as the Transformer model.
- Output: Binary classification (ECM-related or not).

The Transformer model used the Adam optimizer with a learning rate of 5e-5 and a batch size of 16. The LSTM model used the Adam optimizer with a learning rate of 1e-4 and a batch size of 128. Both baseline models were trained on the full training dataset for 15 epochs.

The training parameters for the baseline models (epochs, batch size, and Adam optimizer with its learning rate) were empirically determined, also taking into account the practical limitations of the GPU environment, especially given the comparatively slower training speed of the LSTM model. These settings were found to facilitate effective convergence and robust performance for these deep learning architectures on the protein sequence classification task. Similar to EPOP, while a formal hyperparameter optimization sweep was not performed, these choices were informed by preliminary experimentation to ensure satisfactory results and reproducibility.

## Prediction and functional analysis of POP-associated ECM proteins

We applied the EPOP model to predict the ECM-relatedness of peptides identified in the POP proteomic dataset. Our analysis pipeline systematically processed this data to yield a high-confidence list of POP-associated ECM proteins, which then underwent further functional analysis.

Our analysis pipeline consisted of the following steps:

1. Peptide-level Prediction: The trained EPOP model was applied to each peptide sequence identified in the PRIDE dataset (PXD011467), yielding a predicted probability of ECM-relatedness.
2. Threshold Application: Peptides with predicted probabilities greater than 0.168 were selected as high-confidence candidates. This threshold was previously optimized on our validation dataset to maximize the F1-score, ensuring a balanced consideration of both precision and recall for ECM classification.
3. Protein Identification: A multi-step process was employed to map these high-confidence peptides to their corresponding proteins and consolidate the results.
   - Peptide-to-Protein Mapping: Initial associations between PSMs and protein accessions (e.g., Ensembl IDs) were extracted directly from the raw MaxQuant processed XML data.
   - Peptide Deduplication and Consolidation: Redundant peptide entries were deduplicated, representing each unique peptide by its best EPOP score. All associated protein accessions for multi-mapping peptides were retained and consolidated.
   - Ensembl to UniProt Mapping: Consolidated protein accessions (primarily Ensembl IDs) were mapped to UniProtKB entries using the UniProt ID mapping service, facilitating retrieval of standardized annotations.
   - UniProt Annotation Retrieval: Detailed functional annotations for all mapped UniProt proteins were retrieved from the UniProt REST API.
   - Protein-level Aggregation: Peptide-level predictions and UniProt annotations were aggregated. For each unique protein (UniProt ID), the maximum EPOP score among its corresponding peptides was assigned as its representative ECM probability, yielding a final protein-centric table.

4. Functional Enrichment Analysis: To gain deeper biological insights into the identified POP-associated ECM proteins, functional enrichment analysis was performed using Enrichr [57]. Genes corresponding to the predicted proteins were analyzed for overrepresentation in established biological pathways and processes, leveraging various public databases including GO, KEGG [55], and Reactome [56]. This analysis was performed on the full set of identified proteins, with subsequent interpretation focusing on terms and pathways explicitly related to ECM.

5. Literature-based Validation: To assess the biological relevance and potential novelty of our predicted ECM proteins, we conducted a comprehensive literature review. This process involved manual searches using Google Scholar and PubMed, combining general terms such as "pelvic organ prolapse", "extracellular matrix", "collagen", "elastin", "fibronectin", and "tenascin" with specific protein names or UniProt identifiers derived from our proteomic predictions. For each protein, we carefully examined the retrieved publications to determine whether there was clear evidence of a functional role or direct association with POP. This systematic review allowed us to classify our predicted ECM proteins into those previously reported in POP literature and novel candidates for the condition.

This multi-faceted comparison allowed us to assess both the reproducibility of our findings and the potential of our approach to uncover novel POP-associated ECM proteins.

## Model interpretability analysis

To gain insights into the sequence features driving EPOP's predictions, we conducted three complementary analyses on high-confidence ECM-related peptides:

1. Single amino acid importance:
   - Method: Iteratively masked each position in the peptide sequence.
   - Metric: Change in predicted probability compared to original sequence.
   - Visualization: Line plots of importance scores.
2. Amino acid substitution impact:
   - Method: For each position, we systematically tested all possible single amino acid substitutions.
   - Metric: Change in predicted probability for each substitution.
   - Visualization: Heatmaps of substitution effects.
3. Sliding window analysis:
   - Method: Extracted fixed-length subsequences and obtained predictions for each window.
   - Metric: Predicted ECM-relatedness probability for each window.
   - Visualization: Score curves along peptide sequences.

These analyses provide a comprehensive view of the sequence determinants of ECM function, offering valuable insights for future experimental studies and potential therapeutic interventions.

By integrating cutting-edge AI techniques with rigorous biological validation and interpretability analyses, our methodology offers a powerful framework for uncovering novel insights into the molecular basis of POP and other ECM-related disorders.

## Results

### Performance evaluation of ECM function prediction models

The performance of the EPOP model on the ECM function prediction task was comprehensively evaluated using an independent test set of 16,000 samples (8,000 positive and 8,000 negative) that did not overlap with the training data.

EPOP achieved outstanding performance on the test set, with an accuracy of 99.40%, precision of 99.50%, recall of 99.30%, and an F1-score of 99.40%. These results demonstrate the model's robust ability to distinguish between ECM-related and non-ECM proteins with extremely high accuracy and sensitivity, showcasing its strong generalization ability.

To further assess the superiority of EPOP, we compared its performance with two baseline models: a Transformer model and an LSTM model. As shown in Table 1, EPOP significantly outperformed both baseline models across all evaluation metrics. The EPOP model improves accuracy by 10.81% compared to the Transformer model and by 21.71% compared to the LSTM model, highlighting the advantages of leveraging pretrained protein language models in capturing evolutionary information and structural patterns from sequences.

A more detailed evaluation, particularly focusing on the balance between precision and recall, is provided by the F1-score. As seen in Table 1, EPOP (ESM-2) achieved an F1-score of 99.40%, indicating an excellent trade-off between precision (99.50%) and recall (99.30%). In contrast, the Transformer's F1-score was 87.74%, reflecting a disparity between its high precision (94.78%) and moderate recall (81.67%), while the LSTM, despite achieving a recall of 84.15%, showed lower precision (74.52%) leading to an F1-score of only 79.04%, suggesting a higher false positive rate. This comprehensive analysis demonstrates EPOP's superior ability to maintain high performance across multiple critical metrics compared to baseline methods.

To provide a comprehensive visualization of EPOP's performance characteristics, we present a figure encompassing key evaluation aspects. This includes the precision-recall (PR) curve (Fig 4A) and the receiver operating characteristic (ROC) curve (Fig 4B) at different probability thresholds. EPOP achieved an area under the PR curve (AUPRC) of 0.999 and an area under the ROC curve (AUROC) of 0.999, demonstrating near-perfect performance. These metrics confirm EPOP's outstanding ability to identify ECM-related proteins, maintaining high accuracy and sensitivity even in the presence of imbalanced positive and negative sample distributions.

Furthermore, Fig 4C illustrates the distinct distribution of EPOP's prediction scores for both ECM and non-ECM proteins on the test set. The model exhibited a clear separation of scores, with positive (ECM) class scores averaging 0.992 (median: 1.000) and negative (non-ECM) class scores averaging 0.005 (median: 0.000). This pronounced distinction, quantified by a score separation of 0.987 (positive-negative), underscores the model's high predictive confidence. Fig 4D presents the confusion matrix at the optimal classification threshold of 0.168. At this threshold, the model achieved high True Positives and True Negatives, alongside

**Table 1**. **Performance comparison of ECM function prediction models on the test set.**

| Model | Accuracy (%) | Precision (%) | Recall (%) | F1-score (%) |
|---|---|---|---|---|
| EPOP (ESM-2) | 99.40 | 99.50 | 99.30 | 99.40 |
| Transformer | 88.59 | 94.78 | 81.67 | 87.74 |
| LSTM | 77.69 | 74.52 | 84.15 | 79.04 |

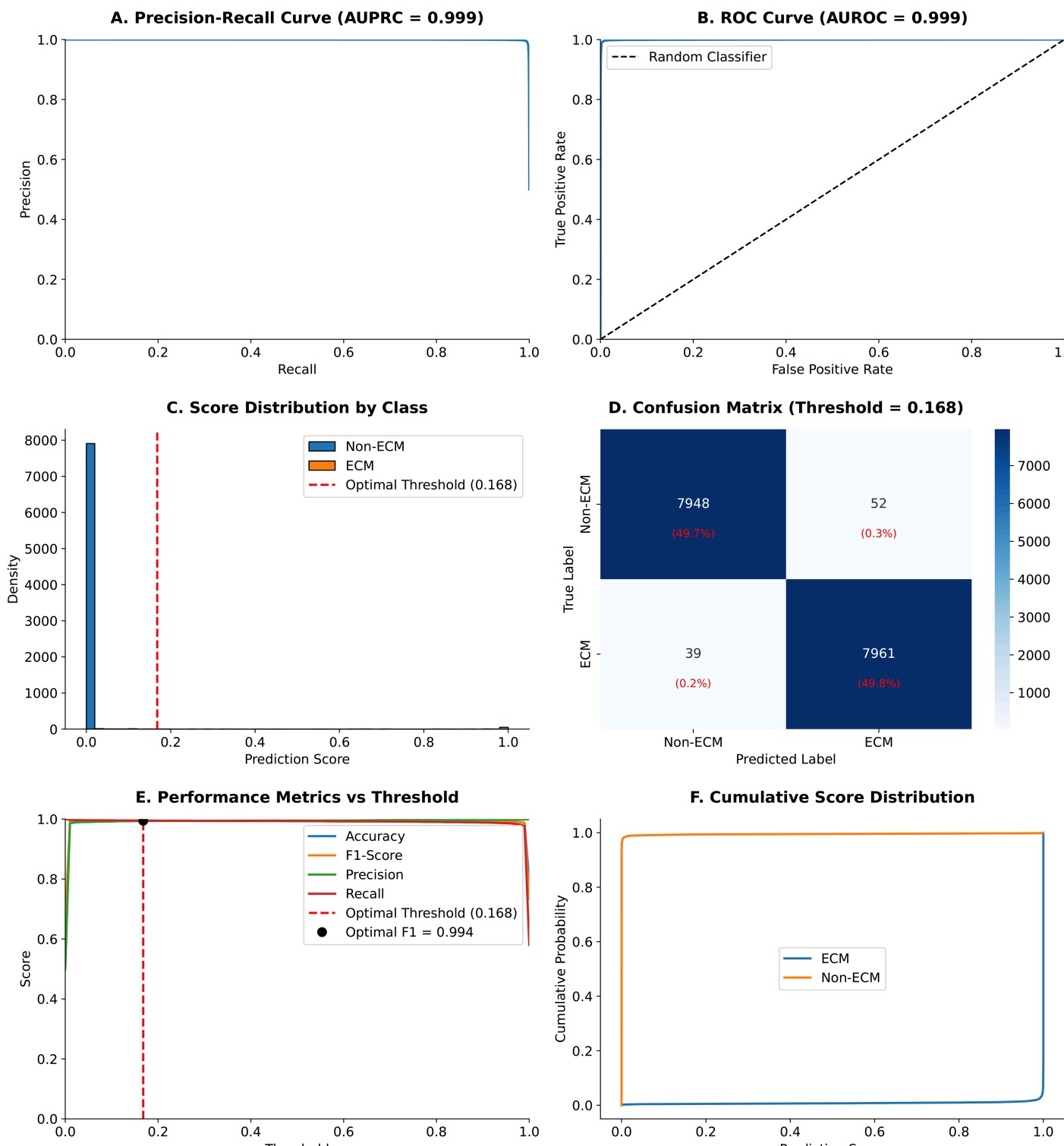

**Fig 4. This figure presents a detailed evaluation of EPOP's performance on the test set, combining core classification metrics with a thorough analysis of threshold effects. A:** Precision-Recall curve (AUPRC = 0.999) demonstrating near-perfect performance with high precision maintained across most recall values. **B:** ROC curve (AUROC = 0.999) illustrating excellent classification performance, substantially outperforming random classification. **C:** Score distribution by class, showing distinct prediction score distributions for ECM and non-ECM proteins. **D:** Confusion matrix at the optimal classification threshold (0.168), providing a detailed breakdown of classification outcomes. **E:** Performance metrics (Accuracy, F1-score, Precision, Recall) across varying classification thresholds, highlighting the optimal threshold (0.168) that maximizes F1-score. **F:** Cumulative score distributions for ECM and non-ECM proteins, visually demonstrating the model's high confidence and clear class separation.

very low False Positives and False Negatives, confirming its exceptional accuracy in classifying both positive and negative instances.

To complement the primary performance metrics and guide practical application, we further analyzed model performance across varying classification thresholds and examined the cumulative score distributions. Fig 4E illustrates how accuracy, precision, recall, and F1-score vary across different probability thresholds, highlighting the inherent trade-offs. An optimal threshold of 0.168 was identified, which maximizes the F1-score. Correspondingly, Fig 4F displays the cumulative distribution functions (CDFs) for both ECM and non-ECM scores, further emphasizing the clear separation in prediction confidence and reinforcing the choice of optimal threshold.

## Prediction results of POP-associated ECM proteins

The EPOP model was applied to mass spectrometry-based proteomic data from POP patients and healthy controls (PRIDE: PXD011467) to predict the ECM-relatedness of identified peptides. High-confidence peptides with predicted probabilities above 0.168 were selected, a threshold optimized for maximizing F1-score. These peptides were subsequently mapped to proteins through a multi-step pipeline, consolidating unique UniProt entries.

This pipeline identified 63 unique high-confidence peptides. When aggregated to the protein level, these peptides yielded a final set of 40 unique, reviewed UniProt proteins confidently predicted by EPOP as ECM proteins. The distribution of EPOP prediction confidence for these 40 identified proteins is visually presented in Fig 5B, showcasing a strong bias towards high confidence scores. The mean EPOP score for these proteins was 0.451, with 6 proteins achieving a score of 0.8 or higher. These identified ECM proteins offer crucial insights into the molecular landscape of POP within the context of patient samples.

To biologically validate EPOP's predictions, we conducted a comprehensive functional enrichment analysis on the 40 identified proteins using Enrichr. As anticipated, these proteins were overwhelmingly enriched in GO terms and biological pathways directly related to the extracellular matrix. The most compelling evidence emerged from the GO Biological

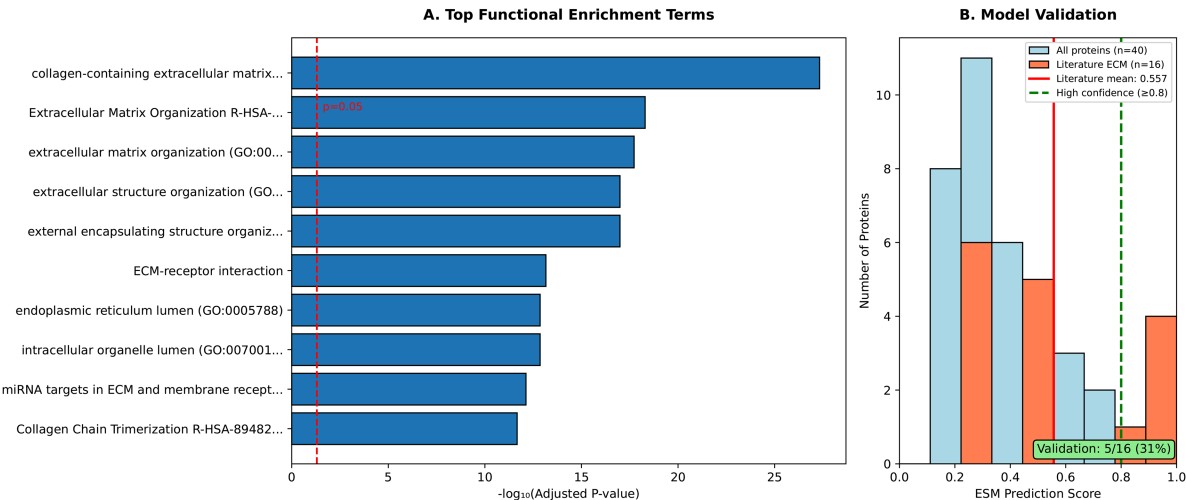

**Fig 5. Key aspects of EPOP's protein identification and validation. A:** Top functional enrichment terms identified among the 40 predicted ECM proteins. **B:** Distribution of EPOP prediction confidence for the identified 40 proteins, highlighting the confidence levels of both literature-supported and novel candidate proteins.

Process 2021 database, where "extracellular matrix organization" (GO:0031012) ranked as the top enriched term with an exceptionally low Adjusted P-value of 1.88e-18. This profound enrichment strongly confirms the core identity of our predicted proteins as central to ECM dynamics.

Furthermore, other highly significant and biologically pertinent enrichments were consistently observed across various databases. For instance, GO Cellular Component 2021 highlighted "collagen-containing extracellular matrix" (GO:0062023, Adjusted P-value: 4.72e-28) and "endoplasmic reticulum lumen" (GO:0005788), reflecting the synthesis and localization of ECM components. Pathway analyses further solidified these findings: KEGG 2021 Human showed strong enrichment in "ECM-receptor interaction" and "Focal adhesion", underscoring the critical interplay between cells and their surrounding matrix. Similarly, Reactome 2022 terms like "Extracellular Matrix Organization", "Collagen Chain Trimerization", and "Collagen Biosynthesis And Modifying Enzymes" were highly significant, detailing the intricate processes of collagen formation and assembly. WikiPathways 2019 Human also pointed to "miRNA targets in ECM and membrane receptors" and "Focal Adhesion" pathways. The comprehensive results of these enrichment analyses are summarized in Figure 5A, visually demonstrating the extensive network of ECM-related functions and pathways represented by our identified proteins. In total, 148 significant terms were found across all queried databases, with 15 explicitly recognized as ECM-related, providing robust biological validation for EPOP's predictions.

We further cross-validated our identified ECM proteins against existing literature concerning POP pathogenesis. Our comprehensive set of 40 predicted proteins was categorized based on their prior association with POP in published studies (Tables 2 and 3).

Table 2 summarizes 16 proteins previously investigated in the context of POP, revealing diverse associations including various expression patterns, structural alterations, or observations of no significant change in affected pelvic tissues. This set largely comprises fundamental ECM components and key regulatory molecules. Our model successfully identified several collagen chains, which are critical for tissue mechanical integrity [64–66]. Literature reports varied expression profiles for these collagens, ranging from downregulation to no significant change. Such findings underscore the complex and heterogeneous nature of

**Table 2. Cross-validation of literature can support proteins.**

| UniProt ID | Protein Name | Expression in POP |
|---|---|---|
| P12111 | Collagen alpha-3(VI) chain | Different expression patterns [58] |
| P02458 | Collagen alpha-1(II) chain | Different expression patterns [58] |
| Q99715 | Collagen alpha-1(XII) chain | Extensive changes [58] |
| Q15582 | Transforming growth factor-beta-induced protein ig-h3 | Extensive changes [58] |
| P98160 | Basement membrane-specific heparan sulfate proteoglycan core protein | Extensive changes [58] |
| Q05707 | Collagen alpha-1(XIV) chain | Differentially expressed [58,63] |
| P22105 | Tenascin-X | Differentially expressed [63] |
| P12109 | Collagen alpha-1(VI) chain | Downregulated [58,59] |
| P04004 | Vitronectin | Downregulated [59] |
| Q06828 | Fibromodulin | Downregulated [63] |
| P02751 | Fibronectin | Upregulated [58,63] |
| P24821 | Tenascin | Upregulated [59,60,63] |
| Q12805 | EGF-containing fibulin-like | Significant enrichment [61,62] |
| P08123 | Collagen alpha-2(I) chain | No significant change [58,59,63] |
| P08572 | Collagen alpha-2(IV) chain | No significant change [59] |
| P05997 | Collagen alpha-2(V) chain | No significant change [59] |

**Table 3. Proteins not explicitly associated with POP in the provided literature.** Functional annotations were adapted from UniProtKB [30].

| UniProt ID | Gene | Function |
|---|---|---|
| Q9HDC9 | APMAP | Involved in adipocyte differentiation and related to fat metabolism, which may indirectly influence tissue composition. |
| Q15063 | POSTN | Matrisome-associated protein promoting cell adhesion and ECM restructuring, enhancing collagen fibril formation and wound repair. |
| Q14112 | NID2 | Cell adhesion glycoprotein, widely distributed in basement membranes, binds to collagens I and IV, perlecan, and laminin 1, likely playing a role in cell-extracellular matrix interactions. |
| P62820 | RAB1A | Key regulator of intracellular membrane trafficking to the cell surface, playing a role in cell adhesion and migration through protein transport. |
| P55290 | CDH13 | Calcium-dependent cell adhesion protein connecting cells, potentially contributing to cell type sorting. |
| P46939 | UTRN | Primarily involved in anchoring the cytoskeleton to the plasma membrane. |
| P43121 | MCAM | Cell adhesion glycoprotein mediating endothelial monolayer cohesion in vascular tissue, relevant for cell-ECM interactions in vascular remodeling. |
| P26038 | MSN | Connects actin cytoskeleton to plasma membrane, regulating cell shape and membrane transport, impacting cellular structural integrity. |
| P25092 | GUCY2C | Guanylyl cyclase involved in cGMP synthesis, a signaling molecule. |
| P23946 | CMA1 | Secreted protease of mast cells, implicated in extracellular matrix degradation and inflammation. |
| P06681 | C2 | Precursor in the complement pathway, mediating pathogen breakdown and adaptive immune system strengthening, relevant for inflammatory responses. |
| P02768 | ALB | Main function is blood colloidal osmotic pressure regulation and transport, indirectly influencing tissue fluid balance. |
| P02749 | APOH | Binds negatively charged substances like phospholipids, potentially influencing blood coagulation. |
| Q9UBG3 | CRNN | Promotes cell proliferation and cell cycle progression, regulating proliferation via inflammatory cytokine responses. |
| Q9H0U4 | RAB1B | Key regulator of intracellular membrane trafficking for vesicle formation and fusion. |
| P02679 | FGG | Forms insoluble fibrin matrix, crucial for hemostasis and early wound repair, stabilizing lesions and guiding cell migration. |
| P00751 | CFB | A precursor of the catalytic component of the alternative complement pathway's C3 and C5 convertase complexes, CFB leads to pathogen phagocytosis and breakdown, and strengthens the adaptive immune system upon cleavage by CFD. |
| P00746 | CFD | A serine protease that initiates the alternative pathway of the complement system by cleaving CFB, leading to pathogen phagocytosis and breakdown, and strengthening the adaptive immune system. |
| P00505 | GOT2 | Catalyzes L-kynurenine transamination, important for NAD(H) redox balance, amino acid metabolism, and fatty acid uptake. |
| O00159 | MYO1C | Actin-based motor molecule for intracellular movements, linking actin cytoskeleton to membranes and regulating vesicular transport. |
| Q9UGT4 | SUSD2 | Putative cytokine receptor with tumor suppressor activity, may regulate cell proliferation. |
| P01024 | C3 | Complement C3 is a central component of the complement system, contributing to pathogen breakdown and immune response, processes that are indirectly linked to inflammation and tissue remodeling within the extracellular matrix. |
| Q9Y6Q9 | NCOA3 | A nuclear receptor coactivator that stimulates hormone-dependent transcriptional activities and participates in the NF-kappa-B pathway. |
| P08294 | SOD3 | Protect the extracellular space from toxic effect of reactive oxygen intermediates by converting superoxide radicals into hydrogen peroxide and oxygen. |

collagen remodeling in POP, where specific collagen subtypes exhibit distinct responses to disease progression [64–66]. Furthermore, our predictions include Fibronectin (P02751) and Tenascin (P24821), consistently reported as upregulated, and Vitronectin (P04004) and Fibromodulin (Q06828), which showed downregulation. These proteins are known to be involved in tissue repair, cell adhesion, and matrix organization [66,73]. The varied expression patterns of these non-collagenous ECM proteins suggest a dynamic and intricate process of tissue repair and compensatory responses to altered biomechanics in POP [66,73,74]. The accurate identification of these established ECM players, encompassing a spectrum of reported involvements, affirms our model's robust capacity to recognize proteins central to POP pathogenesis and validates its biological relevance [64–66].

Table 3 presents 24 proteins that, to the best of our knowledge, have not been explicitly associated with POP in the current literature. Nevertheless, their annotated biological functions, retrieved from UniProt, suggest plausible involvement in ECM integrity, cell adhesion, tissue remodeling, immune modulation, or inflammation, positioning them as promising novel candidates for further POP-related investigation [30].

Notably, several proteins are directly implicated in ECM architecture or cell–matrix interactions [73]. For example, POSTN (Q15063) is a well-characterized matrisome-associated protein. Research indicates that periostin is associated with extracellular matrix remodeling during wound repair, where it is linked to enhanced cell adhesion and promotes fibrogenic differentiation [67,73,74]. These functions are highly pertinent to pelvic floor tissue remodeling. Similarly, NID2 (Q14112), a basement membrane glycoprotein, profoundly influences matrix cohesion through its interactions with multiple ECM components like collagens I, IV, and laminin, suggesting a central scaffolding role in matrix cohesion [73,74]. MCAM (P43121) and CDH13 (P55290) further contribute to crucial cell–cell and cell–ECM adhesion processes, which are fundamental for maintaining tissue structural integrity [75].

In addition, several proteins participate in inflammation and immune-related matrix remodeling. For instance, CMA1 (P23946), a mast cell-derived protease, is directly implicated in ECM degradation and the regulation of inflammatory responses, with studies showing its extensive effects on ECM remodeling, including the suppression of ECM-related genes and degradation of fibronectin [69]. Furthermore, core components of the complement cascade, including C3 (P01024), CFB (P00751), and CFD (P00746), modulate immune responses and are increasingly recognized for their involvement in tissue remodeling processes [68].

Proteins involved in oxidative stress response or vesicle transport also emerge as functionally relevant. SOD3 (P08294), for instance, provides critical defense against oxidative matrix damage in inflamed tissues by neutralizing extracellular reactive oxygen species [70]. Meanwhile, RAB1A (P62820) and RAB1B (Q9H0U4), though classically recognized as intracellular trafficking regulators, are involved in protein transport pathways essential for secretion and membrane dynamics, thereby indirectly influencing cell adhesion and migration [71,72].

Other notable identified proteins encompass diverse functions that may contribute to tissue homeostasis. UTRN (P46939) and MSN (P26038) are actin-binding proteins crucial for anchoring the cytoskeleton to the plasma membrane, impacting cellular structural integrity [77,78]. GUCY2C (P25092), a guanylyl cyclase, plays signaling roles in intestinal epithelial cells by orchestrating intestinal homeostatic mechanisms [76]. Even metabolic enzymes such as GOT2 (P00505) and ALB (P02768), while not canonical ECM components, may exert systemic effects on matrix composition through redox or fluid balance regulation [79,80].

Collectively, the identification of these 24 proteins, many with functions related to the extracellular matrix, immune modulation, or structural support that have not yet been directly studied in the context of POP, demonstrates the unique potential of the EPOP model to reveal biologically relevant but previously unrecognized molecular candidates. These results provide a valuable foundation for future mechanistic investigations and the discovery of novel biomarkers in POP research beyond currently known associations.

## Model interpretability analysis

To gain deeper insights into the sequence features driving EPOP's predictions, we conducted a series of complementary interpretability analyses on 16 literature-validated ECM proteins identified in our study (listed in Table 2). These analyses aimed to pinpoint critical amino

acids, mutation-sensitive positions, and functionally important regions within their peptide sequences. We present the detailed results for the Collagen Alpha-1(VI) chain (P12109) as a representative example, given its high confidence score (0.9974) and established relevance in ECM (Fig 6).

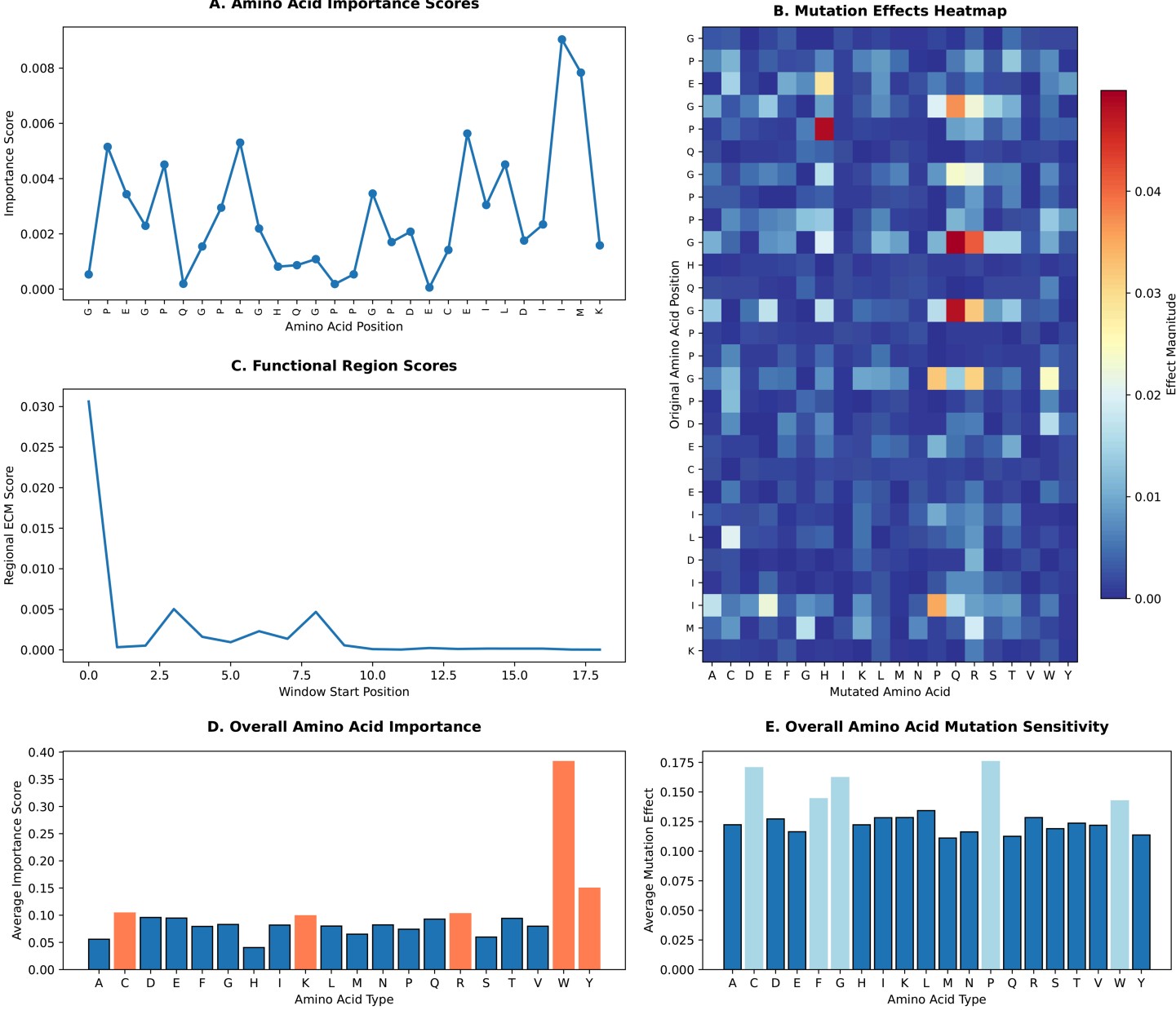

**Fig 6. Interpretability Analysis of Collagen Alpha-1(VI) Chain (P12109) and Aggregated Insights Across Multiple ECM Proteins. A:** Amino acid importance score plot showing varying contributions of different residues, with peaks indicating highly important positions. **B:** Mutation effects heatmap displaying the impact of amino acid substitutions at each position, with color intensity representing effect magnitude (blue to red scale). **C:** Functional region score plot showing the prediction scores across different sequence windows, revealing regions of particular functional importance. **D:** Overall Amino Acid Importance. Average importance score for each amino acid type, aggregated across all positions in all 16 proteins. **E:** Overall Amino Acid Mutation Sensitivity. Average mutation effect for each amino acid type, aggregated across all positions in all 16 proteins.

**Single amino acid importance analysis.** We systematically evaluated the importance of each amino acid at every position for predicting protein function. By iteratively masking individual amino acids in the sequence and observing the resulting change in predicted ECM probability, we quantified the contribution of each residue to the model's discrimination.

For the representative Collagen Alpha-1(VI) Chain (P12109), the results (Fig 6A) showed varying importance across its sequence. Notably, residues such as Isoleucine (I) at position 26 (importance: 0.0090), Methionine (M) at position 27 (importance: 0.0078), and Glutamic acid (E) at position 21 (importance: 0.0056) exhibited consistently higher importance scores. Other impactful residues included Proline (P) at position 9 (importance: 0.0053) and Proline (P) at position 2 (importance: 0.0051). This suggests these residues, even those outside the classical Gly-X-Y motif, contribute significantly to the overall ECM functional prediction.

While previous literature highlights Gly-X-Y repeats as critical for collagen's triple-helical structure where Gly residues (e.g., Glycine at position 10) are essential [38,39], our analysis also reveals the importance of other residues and regions (e.g., positions 26-27), potentially reflecting nuances in ECM protein folding or interactions learned by the model.

**Amino acid mutation effect analysis.** To explore the sensitivity of critical positions to sequence alterations and their role in maintaining protein function, we systematically evaluated the impact of substituting each amino acid at every position with all possible alternative amino acids on the prediction results. The mutation effect heatmap for P12109 is presented in Fig 6B.

The analysis revealed that at key positions, even single amino acid substitutions could significantly alter the predicted ECM probability. For instance, at position 10, a Glycine (G) to Glutamine (Q) mutation (G→Q) caused the largest decrease in ECM prediction probability (effect: 0.0499), followed closely by Glycine to Arginine (G→R) (effect: 0.0411). Similarly, a Proline (P) at position 5 mutating to Histidine (H) (P→H) showed a strong effect (0.0485).

These findings underscore the critical conservation of residues like Glycine and Proline, which are essential for the structural integrity and stability of collagen's triple helix [40,41]. This alignment with previous site-directed mutagenesis and functional studies [42] demonstrates that the model accurately captures biologically relevant mutational sensitivities that impact collagen folding, assembly, and ECM function.

**Sliding window analysis.** To investigate the contribution of local regions to overall protein function, we employed a sliding window approach to scan the peptide sequences and examine the functional prediction scores of different windows. The sliding window score curve for sequence P12109 is shown in Fig 6C.

The results revealed specific regions with consistently high prediction scores, indicating areas of particular functional importance. For P12109, the region from positions 1-10 (GPEGPQGPPG) exhibited the highest regional ECM score (0.0306), suggesting it carries key sequence features. Further comparison of these high-scoring regions with resolved collagen protein structures in the PDB database indicates that they frequently correspond to repeating motifs involved in forming the collagen triple-helix structure, as well as regions containing critical functional sites such as cleavage sites and ligand-binding sites [43].

**Aggregated interpretability insights across multiple ECM proteins.** Beyond the detailed single-protein analysis, we performed an aggregated interpretability analysis across all 16 literature-validated ECM proteins to identify common sequence determinants influencing EPOP's predictions (Fig 6).

As shown in Fig 6D), the aggregated analysis of single amino acid importance revealed that Tryptophan (W) consistently held the highest average importance score (0.3825) across the 16 proteins, significantly higher than other amino acids. This highlights Tryptophan's critical

role in ECM protein function or recognition. Other highly important amino acids included Tyrosine (Y) (0.1497), Cysteine (C) (0.1041), and Arginine (R) (0.1027). Interestingly, while Glycine (G) is fundamental to collagen structure, its average importance (0.0830) was lower than these specific residues, suggesting that while indispensable for core structure, the model might assign higher discriminative power to rarer or more functionally specialized amino acids.

Also, as shown in Fig 6E), the analysis of mutation effects revealed that Proline (P) exhibited the highest overall average mutation sensitivity (0.1757), meaning that substitutions involving Proline (either as the original or mutated amino acid) generally caused the largest changes in prediction probability. Other highly sensitive amino acids included Cysteine (C) (0.1705), Glycine (G) (0.1621), and Phenylalanine (F) (0.1443). The high sensitivity of Proline and Glycine aligns with their crucial structural roles in collagen. The sensitivity of Cysteine suggests the importance of disulfide bonds in many ECM proteins, while Phenylalanine's impact may relate to hydrophobic interactions critical for protein folding and stability. These aggregated findings provide a broader understanding of sequence determinants universally recognized by EPOP across various ECM proteins.

These results not only demonstrate the potential of machine learning methods in revealing protein sequence-function relationships but also provide valuable directions and clues for experimental research and drug development in the context of POP and other ECM-related disorders.

## Discussion

### Implications for understanding POP pathogenesis and ECM roles

Our study represents a significant advancement in unraveling the molecular underpinnings of pelvic organ prolapse (POP), a condition that affects millions of women worldwide [22]. Using the power of advanced AI techniques, specifically the ESM-2 protein language model, we have developed EPOP, a highly accurate tool for predicting and characterizing POP-associated ECM proteins.

The extracellular matrix (ECM) plays a crucial role in maintaining the structural integrity and function of pelvic floor tissues [44]. Our findings provide a more nuanced understanding of ECM alterations in POP, stemming from the identification of 40 unique ECM proteins within patient proteomic samples. This intricate pattern suggests a fundamental disruption of the ECM structural framework affecting the biomechanical properties of pelvic floor tissue. For instance, the literature-reported downregulation of specific collagen chains, such as Collagen alpha-1(VI) chain (P12109) in POP patients, suggests a direct contribution to the comprehensive weakening of the ECM structural framework. Each collagen subtype serves distinct mechanical functions – type I provides tensile strength, type III offers elasticity, and types V and VI regulate fibril organization – indicating that POP involves the multifaceted deterioration of tissue integrity rather than isolated protein defects [45–50]. This aligns with previous studies reporting decreased collagen content in POP tissues [22], and our results further refine this understanding by identifying specific collagen subtypes affected.

Conversely, the reported upregulation of tenascin (P24821) and fibronectin (P02751) in POP patients indicates an active, albeit potentially maladaptive, remodeling process. These matricellular proteins are known to play pivotal roles in tissue repair and cell-ECM interactions [14], suggesting their increased expression may represent a compensatory attempt to stabilize the weakened pelvic floor tissues amidst altered biomechanics. Similarly, the observed downregulation of Vitronectin (P04004) and Fibromodulin (Q06828) in POP tissues highlights further complex dysregulations impacting tissue integrity and repair.

In addition to validating known proteins, our study identified 24 new ECM proteins potentially associated with POP These proteins have not been widely reported in POP-related studies but are known to play important roles in ECM structure and function. For example, POSTN (Q15063) helps organize the extracellular matrix and supports collagen fiber formation and wound healing. NID2 (Q14112) is a key glycoprotein in basement membranes, contributing to matrix stability and cohesion. SOD3 (P08294) protects tissues from oxidative damage, especially in inflamed environments. Also, Some of the newly predicted proteins are involved in immune or inflammatory pathways that affect ECM remodeling. These include CMA1 (P23946), C3 (P01024), CFB (P00751), and CFD (P00746). Such proteins may contribute to matrix degradation through immune-mediated mechanisms. These findings expand our understanding of the molecular changes in POP and provide new targets for future research and biomarker development.

## Advantages and challenges of AI-based disease protein discovery

Our study showcases the immense potential of AI-based methods in advancing our understanding of complex diseases. The EPOP model's ability to predict ECM-related proteins with high accuracy (99.40%) and its superior performance compared to traditional machine learning approaches highlight the power of transfer learning and protein language models in capturing the intricate patterns within protein sequences.

The approach offers numerous advantages for protein discovery in disease contexts. The unbiased nature of our method enables the identification of novel POP-associated proteins beyond those targeted in hypothesis-driven approaches. EPOP's computational efficiency allows rapid screening of entire proteomes, making comprehensive analyses feasible at scales impractical with conventional techniques. The interpretability analyses we implemented provide valuable insights into sequence features driving predictions, bridging the gap between computational outputs and biological understanding. Additionally, by leveraging ESM-2's pre-training on millions of evolutionarily diverse protein sequences, EPOP captures subtle sequence-function relationships that would remain invisible to conventional analysis techniques [7,51].

Despite these strengths, several limitations remain. A major limitation is the absence of direct wet-lab or histological validation for the newly identified ECM protein candidates. Although our computational framework offers strong predictions of disease-associated proteins, their specific biological roles and clinical relevance in POP still require rigorous experimental confirmation.

Another important challenge lies in the construction of the negative training set. Our approach defined non-ECM proteins based on the absence of ECM-related GO annotations, further filtered by SignalP and TMHMM to exclude proteins with signal peptides or transmembrane domains. While this strategy enhances label clarity, it introduces the risk of annotation bias: proteins that are truly ECM-related but lack comprehensive or up-to-date annotations (e.g., newly discovered ECM regulators or atypical secreted factors) may have been mistakenly labeled as negative. Consequently, the model might be penalized during training for correctly identifying such proteins, potentially limiting its ability to generalize to unannotated or unconventional ECM components. This highlights a fundamental challenge in supervised learning based on imperfect biological annotations.

Furthermore, the interpretability analyses, while insightful, are also subject to potential biases originating from both the model and the datasets. The ESM-2 model, pre-trained on a broad protein corpus, may reflect inherent biases from its training data distribution, which could influence the specific sequence features highlighted as important for ECM-relatedness.

Furthermore, biases within our specific datasets, such as potential limitations in UniProt annotations used for model training or the characteristics of the POP patient proteomic samples (e.g., sample size, heterogeneity, technical detection limits), might affect the generalizability and robustness of the interpretability findings. While our analyses offer valuable initial insights into ECM protein characteristics, these inherent data and model biases should be considered when interpreting the precise sequence determinants and warrant further validation in diverse contexts.

Moreover, ESM-2 has inherent limitations when applied to ECM proteins, their high molecular weight, repetitive motifs, and extensive post-translational modifications may not be fully captured by sequence-based models alone, potentially affecting prediction accuracy for certain protein classes.

## Future directions

To further advance our understanding of POP and translate our findings into clinical applications, we propose several key directions for future research.

Initial efforts should focus on experimentally validating the newly identified POP-associated proteins, to determine their biological relevance and clinical potential. We propose a stepwise experimental pipeline combining multiple approaches: (i) immunohistochemical staining of pelvic support tissues from POP patients and healthy controls to assess spatial expression patterns; (ii) quantification of protein levels using Western blot [81] on tissue lysates or biological fluids; (iii) in vitro functional assays using mechanically stressed fibroblasts [82], with knockdown or overexpression of target proteins, to evaluate their role in ECM remodeling; and (iv) in vivo validation using animal models of pelvic floor dysfunction to assess their impact on tissue integrity and disease progression. This multi-level approach will help establish the mechanistic roles of these proteins in POP pathogenesis and evaluate their suitability as biomarkers or therapeutic targets.

Moving beyond individual proteins, we can integrate our proteomic data with transcriptomic and metabolomic profiles from matched POP samples. This multi-omics approach will help us map the regulatory networks controlling ECM remodeling and identify the upstream factors driving the ECM changes we observed. Rather than merely addressing the downstream manifestations of ECM dysfunction, this systems biology perspective could uncover novel intervention points that address the root molecular causes of pelvic floor weakening.

We also can enhance our computational analyses using advanced interpretability methods that can extract biological meaning from our AI model's predictions. By analyzing how specific protein domains and structural motifs influence our model's classification decisions, we expect to uncover previously unrecognized protein interaction networks crucial for maintaining pelvic floor integrity. These computational insights will guide our experimental focus and may reveal unique binding interfaces for targeted drug development.

Also, we see considerable value in applying the EPOP framework to related pelvic floor disorders, particularly stress urinary incontinence. Our preliminary analyses suggest overlapping molecular mechanisms between these conditions. A systematic comparison could identify core pathological processes common to multiple pelvic floor disorders, potentially leading to broad-spectrum therapeutic approaches that address several conditions simultaneously.

Beyond direct applications to closely related pelvic floor disorders, the EPOP framework, built on ESM-2, also holds strong potential for extension to other, more diverse ECM-related disorders, including fibrotic diseases [52], cardiovascular conditions involving ECM remodeling [53], and certain cancers [54]. However, applying the model to different disease contexts or populations requires careful adaptation. ECM structure and molecular mechanisms vary

across diseases, and a model trained on POP data may not generalize directly. Likewise, factors such as genetic background, age, and ethnicity influence ECM composition and disease manifestation, underscoring the need for validation in diverse cohorts. For new applications, the classification head of EPOP should be fine-tuned using disease-specific, high-quality proteomic datasets. Integration of relevant biological features and thorough experimental validation would also be essential to ensure prediction accuracy and clinical relevance.

Longitudinal studies tracking changes in ECM protein expression throughout disease progression could identify early biomarkers for POP risk assessment. Such markers might enable preventive interventions before significant tissue deterioration occurs, particularly in high-risk populations with genetic predispositions or following traumatic childbirth.

The development of targeted therapeutic strategies represents the ultimate translational goal of this research. Approaches might include promoting the expression of downregulated collagens, modulating the activity of matrix metalloproteinases, or enhancing crosslinking of existing ECM components to improve tissue strength and resilience. The integration of these research directions could significantly advance our understanding of POP pathogenesis and lead to novel diagnostic and therapeutic approaches.

Despite these promising avenues, there are several barriers to clinical translation, including the current lack of routine ECM protein testing in clinical practice. However, as personalized medicine advances and the importance of the ECM in various diseases becomes more recognized, the integration of proteomic research in clinical settings could yield significant benefits for millions of women affected by this condition worldwide.

In conclusion, our study demonstrates the power of AI-driven approaches in uncovering novel insights into complex diseases like POP. By combining advanced machine learning techniques with rigorous biological validation, we have not only deepened our understanding of POP pathogenesis but also paved the way for future research into ECM-related disorders. As we continue to refine and expand this approach, we anticipate significant advancements in the diagnosis, prognosis, and treatment of POP.

## Conclusion

In this study, we have demonstrated the remarkable potential of AI-based methods, specifically the ESM-2 protein language model, in predicting POP-associated ECM proteins and elucidating their functional roles in disease pathogenesis. By leveraging transfer learning and fine-tuning the ESM-2 model on expertly curated datasets, we developed EPOP, a highly accurate and efficient model for identifying proteins with high ECM functional relevance from their amino acid sequences.

The application of EPOP to POP patient-derived proteomic data revealed a set of novel POP-associated ECM proteins, potentially playing crucial roles in ECM remodeling and metabolic dysregulation underlying POP pathogenesis. Functional enrichment analyses supported the biological significance of these predicted proteins by highlighting their involvement in key ECM-related processes and pathways.

A comparison of our predicted POP-associated ECM proteins with previous research findings provided strong evidence for their biological relevance. The substantial overlap with known associations reported in previous studies, as well as the identification of novel proteins implicated in other ECM-related disorders, underscores the value of our approach in uncovering new insights into POP pathogenesis.

Our study highlights the immense potential of AI-based methods in advancing our understanding of complex diseases and facilitating the discovery of novel disease-related proteins. The ability of deep learning models to learn rich representations from vast amounts of protein

sequence data has revolutionized the field of protein function prediction. Our fine-tuning approach demonstrates the feasibility and effectiveness of adapting pre-trained language models to specific disease contexts, enabling the identification of novel disease-associated proteins and the exploration of their functional roles.

This promising methodology now needs to be translated into clinical practice, where its potential to enhance diagnostic and therapeutic strategies can be fully realized. Future work should focus on experimental validation of predicted proteins, integration with other omics data, and development of targeted interventions based on these findings. As we continue to refine and expand this approach, we anticipate significant advancements in the diagnosis, prognosis, and treatment of POP, ultimately improving patient outcomes and quality of life.

## Acknowledgments

We extend our thanks to the technical staff at the University College Cork and Munster Technological University for their assistance with data processing and computational resources. Additionally, we acknowledge the open-source community, whose tools and platforms facilitated the development of the EPOP model and the dissemination of our findings.

## Author contributions

**Conceptualization:** Yanlin Mi, Sabin Tabirca.

**Data curation:** Yanlin Mi, Ben Cahill, Venkata V. B. Yallapragada.

**Formal analysis:** Yanlin Mi, Ben Cahill.

**Funding acquisition:** Sabin Tabirca.

**Methodology:** Yanlin Mi, Ben Cahill.

**Project administration:** Yanlin Mi.

**Resources:** Yanlin Mi, Venkata V. B. Yallapragada.

**Software:** Yanlin Mi.

**Supervision:** Barry A. O'Reilly, Sabin Tabirca.

**Validation:** Yanlin Mi, Ben Cahill, Venkata V. B. Yallapragada.

**Visualization:** Yanlin Mi.

**Writing – original draft:** Yanlin Mi.

**Writing – review & editing:** Yanlin Mi, Ben Cahill, Venkata V. B. Yallapragada, Reut Rotem, Barry A. O'Reilly, Sabin Tabirca.

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
