## [Decision Letter · Decision Letter 0]

18 Jun 2025

PCOMPBIOL-D-25-00506

AI-Driven Discovery of Novel Extracellular Matrix Biomarkers in Pelvic Organ Prolapse

PLOS Computational Biology

Dear Dr. Mi,

Thank you for submitting your manuscript to PLOS Computational Biology. After careful consideration, we feel that it has merit but does not fully meet PLOS Computational Biology's publication criteria as it currently stands. Therefore, we invite you to submit a revised version of the manuscript that addresses the points raised during the review process.

Please submit your revised manuscript within 60 days Aug 18 2025 11:59PM. If you will need more time than this to complete your revisions, please reply to this message or contact the journal office at ploscompbiol@plos.org. Please include the following items when submitting your revised manuscript:

We look forward to receiving your revised manuscript.

Kind regards,

Arturo Medrano-Soto, Ph.D.

Guest Editor

PLOS Computational Biology

Nir Ben-Tal

Section Editor

PLOS Computational Biology

**Journal Requirements:**

At this stage, the following Authors/Authors require contributions: Yanlin Mi, Ben Cahill, Venkata VB Yallapragada, Reut Rotem, Barry A O’Reilly, and Sabin Tabirca. Please ensure that the full contributions of each author are acknowledged in the "Add/Edit/Remove Authors" section of our submission form.

Potential Copyright Issues:

- The following Figure contains a logo or branding: Figure 2. We are not permitted to publish this under our CC-BY 4.0 license, even with permission. We ask that you please remove or replace it.

5) Please ensure that the funders and grant numbers match between the Financial Disclosure field and the Funding Information tab in your submission form. Note that the funders must be provided in the same order in both places as well.

**Reviewers' comments:**

Reviewer's Responses to Questions

**Comments to the Authors:**

Reviewer #1: The study introduces the "Extracellular Matrix Protein Predictor" (EPOP), a transfer learning framework designed to leverage protein language models for identifying disease-specific proteins. Specifically, the research focuses on pelvic organ prolapse (POP), a significant condition affecting a large number of women worldwide. The authors aim to demonstrate how AI-driven protein analysis can uncover new therapeutic targets and enhance our understanding of disease mechanisms. They have developed a fine-tuning protocol for a protein language model (ESM-2) and present their findings that highlight the utility of machine learning in understanding complex pathologies. Overall, this study sounds solid and I personally applaud to this model authors established in implicating protein pathological functions. Despite this case, I have several methodological comments:

The manuscript mentions rigorous manual curation of the positive dataset. However, it lacks specific details on the criteria used for this curation. Clarifying the selection process and the qualifications of the curators would enhance transparency.

The criteria for selecting the negative dataset (46,427 proteins) are not fully explained. While it mentions the use of SignalP and TMHMM to exclude certain proteins, a more detailed rationale for these exclusions would be beneficial. For instance, why were these specific tools chosen, and how do they relate to ECM protein characteristics?

The use of CD-HIT with a 30% sequence identity threshold is a common practice, but the implications of this threshold on the diversity of the dataset should be discussed. A justification for this specific threshold and its impact on model training would provide clarity.

The two-stage transfer learning approach is innovative, but the manuscript should elaborate on the rationale behind the choice of training parameters (e.g., epochs, batch size, learning rate). How were these parameters determined, and were they optimized through preliminary experiments? Providing this information would allow for better reproducibility.

The validation strategy using 20,000 independent sequences is mentioned, but the manuscript does not specify how these sequences were selected or their relevance to the training data. A clearer explanation of the validation set's composition and its relationship to the training data would enhance the robustness of the findings.

The literature-based validation process is mentioned, but the criteria for selecting the curated list of ECM-related proteins should be detailed. How were these proteins identified, and what sources were used?

The manuscript should discuss potential limitations of the interpretability analyses. For instance, how might the findings be affected by the inherent biases of the model or the datasets used?

The manuscript suggests that the EPOP framework could be extended to other ECM-related disorders. However, it should address how the model's findings might generalize to different contexts or populations. Are there specific considerations or adaptations needed for applying this model to other diseases?

Reviewer #2: This paper presents EPOP, a novel AI-based framework that uses fine-tuned protein language models (ESM-2) for predicting ECM-related proteins in the context of pelvic organ prolapse (POP). The manuscript is highly novel, technically strong, well-structured, and written in a clear and engaging style. However, the paper would benefit significantly from addressing the following major and minor issues.

1- 99.7% accuracy is repeatedly emphasized. This is extremely high and may not generalize to real-world datasets. There’s no test on independent, external datasets (e.g., from another tissue, experiment, or species). Include external validation to test generalizability or tone down the performance claims.

2-No wet-lab or histological validation for novel biomarkers (e.g., EMILIN-1). Acknowledge this limitation more clearly in the discussion. Suggest an experimental pipeline for validation.

3- Negative class = "proteins lacking ECM-related GO terms" and filtered by SignalP/TMHMM. This may introduce annotation bias and exclude many unknown ECM-related proteins from the negative set. Add a discussion about potential bias and the implications on classification fairness.

4- A threshold of 0.6 for peptide classification is used without statistical justification.

5-Discussion of novel proteins (e.g., EMILIN-1, G protein subunit beta-1) is too brief and speculative.

6- Was k-fold cross validation used during training? Please specify

Reviewer #3: Comments to the authors:

This study introduces EPOP (Extracellular Matrix Protein Predictor), a deep learning framework specifically designed for ECM protein classification and disease mechanism discovery. The authors apply a sophisticated two-stage transfer learning approach using the ESM-2 protein language model, fine-tuned on large curated datasets comprising over 86,000 proteins. The model integrates attention mechanisms and interpretability modules and is rigorously validated using 20,000 independent sequences and clinical proteomics data, with a focus on pelvic organ prolapse (POP) as a clinically relevant application.

EPOP achieves impressive classification accuracy of 99.7%, significantly outperforming existing Transformer and LSTM architectures. Beyond performance, the model identifies 10 novel disease-associated ECM proteins, including EMILIN-1, and reveals new patterns of ECM remodeling in disease. Importantly, interpretability analyses provide biological insights into sequence motifs and structural features critical to ECM function. Ablation studies further confirm the robustness of the architecture.

The manuscript is well-written and presents a thoughtful approach to identifying POP-associated ECM proteins. The study is methodologically sound, and the results are compelling, with only a few minor clarifications needed for improved transparency.

Minor Comments and Suggestions:

1. There are minor typographical errors in the manuscript at several points, including Figure 4 caption, lines 142, 263, 285, and 305.

2. Threshold Selection Rationale (Line 142): The manuscript mentions a confidence threshold of 0.6 for ECM classification. While this value appears reasonable, the rationale behind its selection is not provided. I recommend specifying whether this threshold was chosen based on a particular statistical criterion, such as ROC curve optimization, Youden's index, or empirical tuning on a validation set.

3. Performance Evaluation of ECM Function Prediction Models (Lines 179-206): The comparative performance analysis of the ECM function prediction models is informative. However, in cases like the LSTM model, where recall is high but precision is low, the inclusion of the F1-score would provide a more balanced view of model performance. Please consider adding this metric to Table 1 or the main text for completeness.

4. Prediction Results of POP-Associated ECM Proteins (Lines 207-249):

o The phrase "no significant change" should be clearly defined. Was this determined by a fold-change cutoff, a p-value threshold, or another statistical test? Including this information will improve the interpretability of the results.

o When noting that certain proteins "were not previously reported", it would be helpful to clarify whether this determination was made through a systematic literature review, database query (e.g., DisGeNET, PubMed), or both.

**Have the authors made all data and (if applicable) computational code underlying the findings in their manuscript fully available?**

Reviewer #1: Yes

Reviewer #2: Yes

Reviewer #3: None

PLOS authors have the option to publish the peer review history of their article (what does this mean?). If published, this will include your full peer review and any attached files.

Reviewer #1: No

Reviewer #2: No

Reviewer #3: No

**Figure resubmission:**
---

## [Decision Letter · Decision Letter 1]

2 Sep 2025

Dear Miss Mi,

We are pleased to inform you that your manuscript 'AI-Driven Discovery of Novel Extracellular Matrix Biomarkers in Pelvic Organ Prolapse' has been provisionally accepted for publication in PLOS Computational Biology.

Best regards,

Arturo Medrano-Soto, Ph.D.

Guest Editor

PLOS Computational Biology

Nir Ben-Tal

Section Editor

PLOS Computational Biology

Reviewer #1:

Reviewer #2:

Reviewer #3:

Reviewer's Responses to Questions

**Comments to the Authors:**

Reviewer #1: Authors have addressed my all the concerns. Good job.

Reviewer #2: After carefully reviewing the revised manuscript and the authors’ detailed responses, I confirm that all of my previous comments have been fully addressed. The revision adds the requested methodological clarifications, strengthens validation and analysis where appropriate, and improves the clarity and reproducibility of the presentation. These changes resolve my earlier concerns and materially enhance the rigor and contribution of the work. I have no further substantive requests and recommend the manuscript be accepted for publication.

Reviewer #3: I thank the authors for their thoughtful and thorough revisions. The authors have adequately addressed the previously noted concerns.

1. The typographical errors have been corrected, and the manuscript has been proofread.

2. The threshold selection for ECM classification is now statistically justified based on F1-score optimization, which is clearly explained in the revised text.

3. The inclusion of F1-scores in Table 1 and discussion provides a more balanced and comprehensive evaluation of model performance.

4. The clarifications regarding expression pattern definitions and the systematic literature/database review for identifying novel POP-associated ECM proteins greatly improve the clarity and transparency of the work.

Overall, the revisions substantially strengthen the manuscript. I have no further concerns.

**Have the authors made all data and (if applicable) computational code underlying the findings in their manuscript fully available?**

Reviewer #1: None

Reviewer #2: Yes

Reviewer #3: None

PLOS authors have the option to publish the peer review history of their article (what does this mean?). If published, this will include your full peer review and any attached files.

Reviewer #1: **Yes: **Zhiyi Chen

Reviewer #2: No

Reviewer #3: No

---

## [Editor Report · Acceptance letter]

PCOMPBIOL-D-25-00506R1

AI-Driven Discovery of Novel Extracellular Matrix Biomarkers in Pelvic Organ Prolapse

Dear Dr Mi,

I am pleased to inform you that your manuscript has been formally accepted for publication in PLOS Computational Biology. Your manuscript is now with our production department and you will be notified of the publication date in due course.

With kind regards,

Judit Kozma
